# Clozapine and Pneumonia: Synthesizing the Link by Reviewing Existing Reports—A Systematic Review and Meta-Analysis

**DOI:** 10.3390/medicina60122016

**Published:** 2024-12-06

**Authors:** Victor Zhao, Yiting Gong, Naveen Thomas, Soumitra Das

**Affiliations:** 1Department of Psychiatry, University of Melbourne, Parkville 3052, Australia; Victor.Zhao2@mh.org.au (V.Z.); yitinggong123@gmail.com (Y.G.); 2Department of Psychiatry, Western Health, Footscray 3011, Australia; Naveen.Thomas@wh.org.au

**Keywords:** clozapine, antipsychotics, pneumonia, severe mental illness, vaccination

## Abstract

*Background and Objectives:* Clozapine is a highly effective antipsychotic used for treating treatment-refractory psychotic and mood disorders. However, clozapine also has a serious risk of side effects leading to mortality, particularly its potentiated risk of leading to pneumonia. This review aims to overview the demographic and health-related risk factors leading to pneumonia to better inform risk assessment for clozapine users and to summarise current theories on the mechanisms for clozapine-associated pneumonia. This paper will highlight the need to prioritise pneumococcal vaccination in this population group. *Materials and Method:* We conducted a literary search of five online databases conforming to PRISMA. Our review includes all peer-reviewed papers with original data that discuss clozapine and pneumonia and excludes case reports. Baseline information of participants, pneumonia-related information and information regarding risk factors and mechanisms causing pneumonia were also extracted. *Results:* Clozapine was found to have an increased risk of pneumonia compared to other antipsychotic medications. Factors included comorbidities, higher clozapine dosages, and concurrent use of other antipsychotic medications. Key mechanisms for clozapine-associated pneumonia include clozapine-induced hyper sedation, sialorrhea and neutropoenia. *Conclusions:* While clozapine improves overall mortality for patients, our review confirms clozapine has the highest risk of pneumonia of all antipsychotics. The review also highlights the prevalent underuse of pneumococcal vaccines among clozapine users and the urgent need to increase uptake.

## 1. Introduction

Treatment-refractory schizophrenia (TRS) is defined as failed treatment despite two or more trials of different antipsychotics with sufficient dosages and durations [1]. TRS may cause significant impairment to patients’ social, cognitive, and physical functions, such as unemployment, cardiovascular disease, and relationship difficulties [2,3]. Clozapine is a second-generation antipsychotic (SGA) recognised for its clinical superiority over other antipsychotics in the treatment of various psychotic and mood disorders such as TRS and bipolar disorder (BPAD) [4]. Previous systematic reviews found that clozapine is the most effective antipsychotic for the treatment of positive symptoms in the long and short term [4,5]. This is supported by a Lancet meta-review, which found that clozapine has the greatest improvement in overall symptoms and negative symptoms. There were also strong benefits for positive symptoms compared to other antipsychotics [6].

However, clozapine has an unfavourable haematological and metabolic side effect profile, and there is an increased need for monitoring [7]. Common side effects include sialorrhea, dizziness and headaches [8]. Clozapine is also linked to more serious side effects such as myocarditis, agranulocytosis, metabolic disorders, and pneumonia [8,9]. According to the 2020 World Health Organisation Vigibase data, pneumonia was the greatest specified contributor to clozapine death globally, causing 30% of all cases [9]. Other studies have also indicated that pneumonia is the leading cause of hospitalisations and intensive care admission for clozapine.

Currently, most antipsychotics have been generally found to be associated with an increased risk for pneumonia [10,11]. Of these, clozapine has the highest risk for pneumonia [12], yet the mechanism of action for this risk is currently poorly understood as it is under-appreciated [13]. Studies rarely distinguish between aspiration pneumonia and infective pneumonia, which may have different aetiologies. It is broadly understood that neutropenia, sialorrhea and Parkinsonian features may contribute to pneumonia risk [14]. However, there is conflict around the patient factors contributing to this risk and the underlying pharmacodynamic and immunological mechanisms. Thus, this is the first review to our knowledge that contains the recent large-population studies exploring the comparative pneumonia risk between clozapine users and other antipsychotics. Our review also includes the recent evidence exploring the pharmacodynamic and immunological basis of clozapine-induced pneumonia.

Patients with severe mental illness are known to have the highest risk of developing pneumonia and severe COVID-19 symptoms yet have lower rates of influenza, COVID-19 and pneumococcal vaccination [15,16]. Our review highlights the need to prioritise clozapine users for all vaccinations, particularly pneumococcal vaccination.

The primary outcomes of this review were as follows:-The level of risk of pneumonia in clozapine patients compared to other antipsychotics;-The demographic and physical factors affecting the risk of pneumonia;-Identification of the causes underlying clozapine-associated pneumonia.

## 2. Materials and Methods

### 2.1. Studies Considered and Their Selection

This review was conducted following PRISMA guidelines (Appendix A) by searching Embase (Elsevier Amsterdam, The Netherlands), PubMed (National Institute of Health Bethesda, MD, USA), Scopus (Elsevier Amsterdam, The Netherlands), PsycINFO (American Psychological Association Washington, DC, USA) and MEDLINE (National Institute of Health Bethesda, MD, USA) using keywords related to pneumonia, clozapine and antipsychotics, which were searched in the title, keywords and abstract. A separate search was also conducted using keywords relating to severe mental illness, schizophrenia and vaccination (Table 1). Details of the search strategy are in Appendix A. The following limits were applied to search results: English, humans, and publication year 1990–2023. Results were then uploaded into COVIDENCE software (Melbourne, Australia) to automatically account for duplicates. VZ then screened all titles and abstracts using COVIDENCE, followed by performing a full-text review of the remaining papers. YTG later performed a secondary screening to ensure the accuracy of the search, blinded to VZ’s selection results. Any conflicts that arose were resolved by SD. Duplicates were further filtered out manually. All screening was performed according to the inclusion and exclusion criteria, which were agreed upon by all authors before screening commenced and verified with a preliminary search to ensure adequate sampling (Table 2) and quality, assessed by VZ and YTG using Joanna-Briggs Institute Checklist (Table 3 and Table 4). A PRISMA flowchart was generated by COVIDENCE (Figure 1). The supervisor SD went through this to identify the authenticity of the search strategies. Furthermore, papers were also screened for the following secondary outcomes (Table 2).

### 2.2. Data Extraction

Data extraction was then performed on all included papers by VZ and YTG to collect baseline characteristics of the included studies, demographic data, and data on pneumonia risk compared to other antipsychotics in terms of odds ratios. Any missing data were manually calculated by the authors. Additional data, such as those obtained from the reporting of the type of pneumonia and discussion of mechanisms, were also collected. A data extraction table with pre-specified outcomes was created in consultation with all authors on Excel and used for all extraction. A trial of the table was conducted on 5 papers before data extraction was performed by VZ and YTG independently. Any differences in extraction were referred to SD for consultation and final decision-making. All extracted information is tabulated in Table 5, Table 6 and Table 7.

### 2.3. Statistical Method

Adjusted effect estimates were pooled using a weighted random effects model. Heterogeneity was estimated by using forest plots and the I^2^ statistic [37]. I^2^ was considered low (25%), moderate (50%) or high (75%) [38]. We considered examining potential factors contributing to heterogeneity, such as age, gender, and study design, using subgroup analysis and meta-regression; however, given the limited number of studies in each comparison (seven or fewer studies), such tests would be unlikely to be reliable due to reduced power [39] and were therefore not undertaken. We initially assessed the possibility of publication bias using the funnel plot, adjusted by Egger’s regression test [40]. Additional sensitivity analyses were performed by pooling studies using a random effects model with Knapp–Hartung adjustments [41]. All statistical analyses were performed in R version 4.3.3 statistical packages (R Core Team, 2016) [42].

## 3. Result

A total of 2100 papers were found from the initial search, with 764 duplicates removed. This left 1336 papers for abstract screening. 1172 papers were then removed, which left 164 papers for full-text review. In total, 23 papers with 330,699 participants (clozapine users only) remained after all screening to be included in data extraction. No additional papers were identified from secondary sources. Our meta-analysis included nine papers.

### 3.1. Study Characteristics

The following demographic data were extracted from the studies:-Location of study;-Study format;-Length of time of study.

Of the 23 papers included, 8 papers originated from Taiwan, 5 from the United States of America, 4 from China, 2 from Denmark and 1 from the United Kingdom, Japan, and

New Zealand. A multinational study conducted by the WHO was also included. Of these papers, there were 1 randomised control trial, 3 case–control studies, 6 cohort studies and 13 database or registry review studies. The length of time of study also varied. Registry/database review studies varied from 2 to 50 years, cohort studies varied from 2 years to 6 years, case–control studies varied from 2 years to 24 years, and the randomised control trial lasted for 12 weeks. A summary of study characteristics can be found (Table 5). In total, 8 papers had aims specifically relating to the relationship between clozapine and pneumonia, and 15 papers did not have aims specific to clozapine but included data specific to clozapine.

### 3.2. Demographic Data

The following demographic data were extracted from the studies:-Number of participants taking clozapine;-Sex percentage of clozapine users;-Age of participants;-Comorbidities of patients (reported as Charlson comorbidity index (CCI));-Reason for clozapine use.

The total number of patients in 23 papers varied from 11 participants to over 140,000 participants. Demographic data for patients taking clozapine were not routinely reported in papers. Only 8 papers included clozapine-specific demographic data (Table 6), 15 papers reported the overall demographic data for all participants (Table 6), and 5 papers did not report any demographic data relevant to extraction. Of the eight papers with clozapine, five report on patients with a diagnosis of schizophrenia with a mean age between 38.5 and 57.2 years, with females representing between 33% and 43.3% of the population. Only three of these papers provided a CCI, while four provided only a percentage of each comorbidity found in patient populations, and one did not report any information. In total, 87.1% of clozapine users with schizophrenia had a CCI score of 0–1, and 12.9% had a CCI score of 2 or more.

### 3.3. Prevalence and Incidence of Pneumonia Risk

The prevalence of clozapine-associated pneumonia was described in 10 papers. A large heterogeneity of percentages was identified, the lowest being 1.87% and the highest being 34%, with the mean being 6.18% (Table 6). In addition, Rhode [14] and Wu [21] reported the incidence of pneumonia, reporting 2.14/100 person-years and 2.98/100 person-years, respectively (Table 7). Both papers reported exclusively on schizophrenic patients.

### 3.4. Risk of Pneumonia with Re-Exposure to Clozapine

Hung [34] and Li [35] described the risk of recurrent pneumonia after reintroducing clozapine compared to initial use, reporting odds ratio (OR) = 1.25 [35] and OR = 1.99 [34], respectively (Table 7).

### 3.5. Dose Dependency of Pneumonia Risk

Four papers [12,22,34,35] reported a dose-dependent increase in pneumonia risk with increased clozapine dosages. This is supported by higher incidences of pneumonia among schizophrenic patients compared to bipolar patients, who are usually on lower doses of clozapine. However, Stoeker et al. [32] found that the dose-dependent increase in pneumonia risk is not statistically significant.

### 3.6. Temporal Risk of Clozapine Use

Five papers [12,17,21,29,34] found that the greatest risk for pneumonia in clozapine use was in the first 30 days. Wu compared the risk of pneumonia based on the duration of treatment to discontinuation and found that the OR = 4.00 in the first 30 days decreased to OR = 0.93 after 3 months and OR = 0.71 after 6 months, when it would plateau. Similarly, another paper reported OR = 9.57 compared to that for non-current use [12]. However, another paper [20] found the greatest risk of pneumonia with extended use of clozapine, which was hypothesised due to immunosuppression.

### 3.7. Individual Use Compared to Non-Antipsychotic Use

In total, three papers were identified that reported the odds ratio for risk of pneumonia on clozapine compared to no antipsychotic (OR = 3.73, risk ratio = 2.59 [33]; OR = 4.07 [32]; and OR = 3.21) [25]. These papers had small population sizes of 96, 155 and 120 participants on clozapine, respectively. Yang et al. explored only bipolar patients and excluded those with schizophrenia, while Han only investigated patients with schizophrenia, and Stoeker et al. included patients with schizophrenia, schizoaffective disorder, and bipolar disorder (Table 6).

### 3.8. Comparison of Clozapine to Other Antipsychotics for Pneumonia Risk

Meta-analysis was performed comparing clozapine to SGAs and FGAs separately. Nine studies were included in the analysis for the comparison between clozapine and SGAs for the risk of pneumonia (Figure 2). Clozapine was found to have a statistically significant increased risk compared to other SGAs included in the comparison (OR = 1.65, 95% CI 1.19–2.11). Four studies were included in the analysis between the risk caused by clozapine compared to FGAs (Figure 3), which found an increased risk for clozapine (OR = 1.65, 95% CI 0.55–2.72) but did not reach significance.

### 3.9. Clozapine Use in Conjunction with Other Antipsychotics

Overall, three papers were identified to have discussed the increased risk of the combined use of other antipsychotics with clozapine, with OR = 3.49 [18], OR = 2.49 [20], and OR = 4.80, for clozapine and valproic acid combination [33] compared to single clozapine use.

Kuo et al. [12] reported the OR for individual antipsychotics when combined with clozapine. There was an increased risk for olanzapine (OR = 22.40), quetiapine (OR = 14.76), zotepine (OR = 4.8), risperidone (OR = 7.49) and amisulpride (OR = 21.44), whereas single use of these antipsychotics all had ORs < 1 compared to that for non-current use. However, another paper [25] found an overall decrease in risk with combination antipsychotic therapy.

### 3.10. Mortality from Pneumonia

Mortality from pneumonia was reported in five papers. Two papers reported the results as odds ratios, with OR = 2.17 compared to other antipsychotics [14] and OR = 1.83 when compared to no antipsychotic use [30]. Two other papers [17,43] reported percentages of mortality from pneumonia to be 30% and 23.08%, respectively. Finally, a study by Factor et al. [28] included four deaths from pneumonia among the six patients that contracted pneumonia.

### 3.11. Causes for Pneumonia

In total, 17 papers discussed risk factors for pneumonia in clozapine users. Furthermore, 12 papers published the reporting criteria for pneumonia; of these, 7 papers used the ICD-9 480–486 and 507, 1 paper used ICD10: J12–19, 2 papers reported using chest X-ray/CT scan with clinical parameters, and 1 paper used post-mortem reporting of pneumonia. Eight papers reported the type of pneumonia (hospital-acquired, community-acquired, aspiration pneumonia), 6 of these papers reported hospital-acquired pneumonia, 3 reported aspiration pneumonia, and 3 papers reported community-acquired pneumonia. The summary can be found in Table 7.

### 3.12. Publication Bias

Publication bias was assessed using a funnel plot (Figure 4 and Figure 5) and Egger’s regression test. Analysis of the funnel plots indicates significant heterogeneity in the clozapine comparison with SGA (I^2^ = 87%, *p* < 0.01) and with FGAs (I^2^ = 84%, *p* < 0.01). Visual analysis indicates a positive skew in the clozapine comparison with SGAs, suggesting that papers are potentially missing from the meta-analysis.

## 4. Discussion

Research has previously established that SGAs and, to a lesser degree, FGAs are linked with increased risk for pneumonia [12,33]. Based on current evidence, clozapine appears to have the strongest relationship with pneumonia, which is found generally to be the consensus in most papers (Table 6), except that of Rohde [14], which only found a weak correlation. Our meta-analysis found the same when clozapine was compared to SGAs (OR = 1.65) and FGAs (OR = 1.65). This finding is consistent with previous systematic reviews [7,10,11,44], which all found an increased risk for pneumonia in clozapine-users, with the greatest risk correlated to clozapine. Particularly concerning is that pneumonia appears to be the most common cause of death due to clozapine, which may currently be under-reported and underappreciated [43]. Nose et al. found the upper-risk ratio for clozapine to be OR = 2.35, which correlates to the number needed to harm of 74. This highlights the need for further research into contributing factors to the risk of pneumonia and potential strategies such as pneumococcal vaccination, influenza vaccination and COVID-19 vaccination to protect clozapine users.

### 4.1. Demographic Risk Factors

Demographic factors such as older age, underweight, male sex, being unmarried and lower socioeconomic status are potentially linked with increased risk for pneumonia in schizophrenic, bipolar and dementia patients [14,21,45]. However, there is currently relatively limited research that investigates how demographic features may affect patients taking antipsychotics, particularly clozapine. Risk factors such as advanced age and low body weight remain risk factors for pneumonia in antipsychotic users [23], but it is currently unclear if the risk increases or decreases with antipsychotic use. In terms of clozapine, the currently available evidence indicates that clozapine users are likely to be living alone, be on a pension and have higher CCI levels, which suggests more comorbidities [14] and more use of concurrent medications, which are all risk factors for pneumonia [29,34]. Further, older age, smoking, sex and ethnic background affect clozapine dosing and are pneumonia risk factors [46].

While pneumonia risk increased for both sexes [20], it is currently unclear which sex has a greater risk of pneumonia. It is suggested by Hung [34] that being female leads to greater risk of pneumonia when taking clozapine, but a more recent paper [21] found that male clozapine users were more at risk (hazard ratio = 1.28). This discrepancy is likely due to Hung investigating the risk of recurrent re-exposure, while Wu captured participants who were first recorded in the presentation. Different mechanisms may underlie the risk profile for men and women. One possibility is the increased fat percentage in women, as clozapine may deposit within fat, which can reduce clozapine metabolism and elimination [47]. Increased obesity can also lead to changes in the hepatic metabolism of clozapine [48]. This change is small and usually clinically insignificant but can be clinically relevant in women with large gains in body fat. Another possibility is the higher oestrogen levels in females, which may inhibit CYP1A2 activity [20]. It was also found that GAF score, age of onset of the disease, and CBC/DC ratio were the major factors when determining the risk of pneumonia among women, while psychiatric symptom severity, family history of psychosis and the dosage of antipsychotics were key risk factors for men [20].

A systematic review by Ruan [49] found that Chinese and East Asian people have lower rates of clozapine clearance compared to Caucasians due to lower levels of CYP1A2 activity. De Leon [46] describes that women typically need half the clozapine dosage of their ethnic male counterparts for Asians and Caucasians, and Asian men need half the dosage of their Caucasian counterparts [50]. This not only supports sex being a key determining factor of risk but that there are significant clinical differences in clozapine metabolism based on geographic ancestry and the need for ethnically based clozapine titration. The key risk factor for pneumonia may be clozapine/norclozapine serum levels or changes in clozapine metabolism.

### 4.2. Medical Conditions Causing Clozapine-Related Pneumonia

Medical conditions such as cardiovascular disease, COPD, diabetes, and asthma have been linked to an increase in the risk of pneumonia in the general population [51]. Studies have also found that many of these same comorbidities have been linked to increased risk among patients on antipsychotics [12,29,35]. However, it is uncertain whether the concurrent use of antipsychotics, in particular clozapine, increases the risk caused by these comorbidities. Some papers have found that certain diseases such as COPD and GERD [32] were noticed to be of higher prevalence in the clozapine group who got pneumonia compared to the non-pneumonia group. However, these correlations do not imply causality. A paper by Hung [34] found that old age, concurrent use of medications and comorbidities may have some interaction with clozapine as there was an increase in pneumonia risk by 40%, possibly suggesting that these factors may have a more robust role in impacting the clozapine risk profile compared to other antipsychotics.

Patients with schizophrenia and bipolar have higher rates of smoking, usually have a more sedentary lifestyle, consume more calories, have poorer sleep and are more likely to use alcohol [52]. It is hypothesised that patients with schizophrenia have higher risks for obesity [46]. These lifestyle factors ultimately mean that patients with schizophrenia and bipolar have a higher risk for comorbidities. Patients with psychiatric disease are known to have poor medication adherence, which can also predispose them to a worse prognosis for comorbidities and more frequent hospitalisations [53,54]. Similarly, patients with Parkinson’s disease also have increased comorbidities due to advanced age and functional decline [55]. This is reflected in the literature, where patients on clozapine had a higher average CCI and subsequently were on more medications and show increased frequency for non-psychiatric presentations to the hospital, both of which are known to also increase the risk for pneumonia [53]. Current research indicates that some medications, such as benzodiazepine, corticosteroids and CYP450 enzyme-inhibiting antibiotics, are related to increased risk for pneumonia [21].

It is hypothesised that concurrent use of some of the medications has a synergistic role to clozapine in increasing the risk for pneumonia [33] via receptor modulation [12] or by impeding clozapine metabolism via CYP cytochrome enzymes leading to increased serum clozapine [56], a key risk factor for pneumonia [57]. The mechanisms for these effects are discussed in detail later in this paper. Tobacco is known to be able to induce increased metabolism of clozapine via the hepatic cytochrome enzyme CYP1A2 by binding to aryl hydrocarbon receptors to increase CYP1A2 expression [58]. Hence, clozapine dosages are often increased in clozapine users who smoke tobacco regularly to compensate for the increased metabolism. Sudden cessation of smoking due to hospitalisation or respiratory infection affects the rate at which clozapine is metabolised and may lead to increased serum clozapine levels [29,53]. It is found that very often, there is a “triple whammy” effect leading to hospital-acquired pneumonia [18]. At admission to hospital for an infection, there is a simultaneous cessation of smoking, systemic infection or inflammation, and the prescription of a medication that either inhibits the CYP metabolism of clozapine or weakens the immune system via medications such as corticosteroids [57]. Further, smoking not only predisposes patients to chronic respiratory disease, but it also increases the risk for respiratory infections such as bronchitis, which exposes patients to the dangers of smoking [59].

It is suggested that comorbidities and underlying mental illnesses such as bipolar or schizophrenia disorder may play a more important role in patients than clozapine [53]. Shoretsanitis [53] proposed that the risk of pneumonia is “2/3rd TRS and 1/3rd clozapine” (adapted model from the original paper in Figure 6). TRS leads to increased risk for obesity [60] and smoking [61], which are risk factors for various comorbidities such as cardiovascular disease, chronic obstructive pulmonary disorder, and diabetes [62,63,64]. These factors may both affect clozapine metabolism and, hence, clozapine levels (particularly smoking due to its action in CYP1A2). The combination of increased comorbidities, changing clozapine levels, and poor medication adherence due to TRS all increase the risk for pneumonia. This can be directly fatal or lead to other fatal consequences such as arrhythmia [65]. This is supported by a paper by Villasante-Tezanos [66] which found that the population-attributable risk (PAR) of pneumonia for TRS was 46%, while that for TRS and clozapine was 64%. This implies that clozapine has a PAR of 19%. Hence, it is found that the factors due to TRS itself play a much larger role than factors caused by clozapine action via receptor and immunomodulatory effects. 

### 4.3. Acute and Chronic Clozapine Use

Though clinical data indicate that pneumonia risk peaks in the first 30 days and over long-term use, the pathophysiology behind these occurrences is largely speculative based on available evidence. A study by Kang et al. found that 52/1408 (3.7%) of patients developed pneumonia within the first 8 weeks of clozapine initiation, and about 50% (25/49) developed pneumonia within the first 2 weeks [67]. It was also found during the 8-week period that there were high incidences of inflammation-related adverse events in both the pneumonia and non-pneumonia groups (fever 89.8% vs. 19.6, CRP elevation 73.5% vs. 9.1%, respectively). Symptoms such as orthostatic hypertension are common with clozapine initiation, often due to rapid titration [68]. Hence, the increase in pneumonia risk may be due to clozapine side effects due to overly rapid titration in some patients. It is found that sialorrhea, a risk for aspiration pneumonia, is increased initially and decreases over 4–6 weeks. The mechanism by which clozapine causes sialorrhea is understood to be muscarinic 3 and 4 activity with minor adrenergic contributions [69,70]. It is also hypothesised [71] that there may be a genetic link to pneumonia risk [68] as some differences in the HLA-DQB1 allotype [72] affect the risk for agranulocytosis. It was noted by Leykin et al. [73] that there was a reduction in immune responsiveness of up to 10% with clozapine that was dose-dependent. Indeed, this suggests that initial clozapine does seem to weaken the immune system and alters the sensitivity of peripheral blood mononuclear cells to clozapine, an effect that is transient and disappears by week 6 [74]. In terms of long-term clozapine, it is hypothesised that chronic clozapine use can cause neutrophil depletion and suppress immune responses with chronic clozapine use [74]. The mechanism for neutropenia is largely due to (1) receptor activity mainly via cholinergic and histaminergic pathways, (2) immune regulation via type 1/3 hypersensitivity causing cytokine upregulation and inflammasome activation, and (3) direct toxicity due to toxic metabolites which can damage marrow stromal cells and neutrophils (Figure 7). This is explored in more detail in Section 4.6.1.

### 4.4. Risk of Pneumonia from Patient Diagnosis or Clozapine

Despite the increased risk of pneumonia due to clozapine, it is currently still unclear if the risk of pneumonia is mainly due to clozapine or due to the inherent increased risk of the patient [14]. According to previous meta-analyses, clozapine has an increased risk for pneumonia compared to no drug use (OR = 3.11, 95%CI 2.59–3.74) [10]. However, patients who take clozapine usually have higher CCI [75], have more severe psychotic or mood disorders [20], are more likely to engage in risky behaviour [76], or may have poorer adherence to all medications compared to patients with less severe psychiatric illness or dementia. In most papers, control for pneumonia outcome was not the same patient group before and after clozapine, but often a different patient group, likely with less severe symptoms. This makes disease severity an important confounding factor. Although the absolute risk for pneumonia was similar in patients with a psychiatric reason for clozapine and dementia [11,44], it is noted by Papola 2019 that for pneumonia, there was an increased risk association with antipsychotics in younger populations than older populations. The reasoning behind this result may be that studies in younger populations were conducted in psychiatric patients where higher antipsychotic dosages are used compared to older populations where dosages are usually lower and may be used less frequently [44], or that certain medications used in the elderly, such as anti-Parkinson medications, confer a protective element to risks of pneumonia [26]. It is also possible that due to the innate increased risk in older patients [77], the relative risk attributed to clozapine is then reduced. It is suggested that the severity and range of patient symptoms influence pneumonia risk rather than diagnosis [78]. As the baseline risk and cause of risk for pneumonia differs due to different diagnoses, the risk conferred by clozapine will also differ. Hence, it is important to consider the individual risk profile of the patient—symptoms [78], comorbidities [79], and medication adherence—when considering individual pneumonia risk.

### 4.5. Adverse Outcomes of Clozapine-Associated Pneumonia

Clozapine is known to cause other adverse events, such as myocarditis, ischemic colitis, and pulmonary thromboembolism [80]. In patients with pneumonia, other conditions, such as myocarditis, DRESS (drug rash with eosinophilia and systemic symptoms) syndrome and acute pyelonephritis, can co-occur [67]. Although the correlation between these conditions and clozapine-associated pneumonia is unclear, it is hypothesised that it is driven by Type 1 hypersensitivity, which may also underlie the increased inflammatory responses in these patients. Type 1 hypersensitivity has also been implicated in affecting clozapine metabolism and driving immunological changes leading to leukopenia. These changes increase the risk for infective pneumonia and are shown in Figure 7. In the same study, a patient who developed acute pyelonephritis was found to have higher serum clozapine. This highlights the important interplay between the various adverse events and the importance of carefully monitoring serum clozapine levels and for any side effects.

### 4.6. Possible Mechanisms for Clozapine-Associated Pneumonia

The exact mechanisms for pneumonia in clozapine users have not been determined. However, this paper will discuss the plausibility of some of the prominent theories that may underlie the mechanism for aspiration and infective pneumonia in patients.

#### 4.6.1. Infective Pneumonia

The key cause for infective pneumonia among clozapine users is impaired immune response due to the immunoregulatory effects of clozapine leading to leukopenia. However, the pathophysiological causes behind how clozapine leads to these immunoregulatory changes is not well understood [46]. It is hypothesised that clozapine is an antagonist for histaminergic, cholinergic, muscarinic, and serotonergic receptors, although the affinity for these receptors varies greatly [12,13,18]. According to a paper by Cepaityte [13], clozapine has the strongest action on H1 and M1 receptors and weaker action on dopaminergic and serotonergic receptors. Infective pneumonia has been particularly correlated with medications with anti-muscarinic (M1) activity [12,26,81,82]. Linear regression modelling by Cepaityte [13] found that clozapine had the greatest risk for infective pneumonia, followed by olanzapine and risperidone with the least pneumonia but also a low muscarinic occupancy percentage. Antihistamine action is also a commonly proposed mechanism for infective pneumonia. Linear regression for H1 receptor activity was also performed and yielded similar findings, yet was statistically insignificant due to insufficient power. This implies that anti-muscarinic activity and, to a lesser degree, anti-histaminergic activity are correlated with infective pneumonia risk. By contrast, similar modelling for dopamine found dopamine receptors to have a negative association with pneumonia risk. This is largely consistent with other papers that have not found antidopaminergic side effects to be relatively common [25,32,33], so the dopamine effect is unlikely to be the driver of pneumonia risk. A weak trend between pneumonia and serotonin receptors was found, potentially suggesting a weak effect from serotonin blockage [13]. However, despite clozapine having a similar receptor occupancy to olanzapine and chlorpromazine, the increase in pneumonia risk for clozapine compared to that for these antipsychotics is significant [13], suggesting a multifactorial relationship beyond side effects due to receptor activity. Muscarinic and histaminergic action also have immunoregulatory effects, which may provide additional explanation for this disparity.

Clozapine is also hypothesised to be able to affect the immunoregulation process within the body by modulation of cytokines or via direct toxicity. Clozapine is metabolised into a reactive metabolite, which can join together endogenous proteins, leading to the formation of new antigens (i.e., haptens) [83]. Haptens are small substances that can combine with a larger molecule to lead to an immune response but cannot elicit an immune response by itself. These haptens can lead to type 1 and 3 hypersensitivity [83] and an inflammatory response with increased cytokines (e.g., IL-2, IL-6, IL-1, TNF-alpha) [20] due to the inflammasome production of IL-1 beta [84]. This will lead to the activation of neutrophils and other peripheral blood mononuclear cells and the downregulation of lymphocytes [85]. Clozapine may also affect the thromboxane A2 receptor and the platelet-activating factor, which causes increased capillary permeability at the alveolar level [13]. Over time, the elevation of specific cytokines such as interferon-1 can lead to neutropoenia due to chronic leukocytosis [86] and a decreased production of immunoglobin, leading to insufficient adaptive immune defence [87]. It is further postulated that the innate immune system activates the adaptive immune system to mediate idiopathic drug-induced agranulocytosis via T-cell activation and clonal expansion [84,88]. If haptenisation occurs in key cellular proteins, it can also lead to apoptosis via cell stress, leading to the release of damage-associated molecular patterns, which further aggravates inflammation via inflammasome production [84]. Clozapine metabolites such as nitrenium ions may also be directly cytotoxic to neutrophils, leading to neutrophil depletion [89], as well as leading to the death of marrow stromal cells [90]. Marrow stromal cells are important in maintaining and producing blood cells, such as immune cells. Hence, via chronic inflammatory responses, haptenisation-led apoptosis, and direct toxicity, neutropoenia can be present. However, it is important to note that immunoregulation plays only a partial role in this process, as agranulocytosis or neutropoenia was not observed consistently in clozapine users with pneumonia [29,91].

Further, there is a bidirectional relationship between infection and clozapine (Figure 6) [46]. It is established that clozapine can increase the risk of pneumonia and infection via the above mechanisms. However, infection may also affect clozapine levels [57]. During an active infection, cytochrome p450 is inhibited via cytochrome p450-inhibiting antibiotics and other p450-competitive medications [92] and directly via cytokine action [57,93]. This then leads to a positive feedback loop which overall worsens pneumonia or infection and increases the risk of morbidity [53].

Clozapine is known to have a narrow therapeutic range [92,94], and increases beyond the therapeutic range are associated with significant side effects [46,57]. Hence, it is hypothesised that the most prominent risk factor for clozapine-associated pneumonia is serum clozapine/nor-clozapine levels or changes in the metabolism and metabolite excretion of clozapine [46]. This is because most risk factors ultimately affect clozapine by inhibiting clozapine metabolism and elevating clozapine levels.

#### 4.6.2. Aspiration Pneumonia

Aspiration pneumonia develops secondary to the aspiration of a foreign body or bodily fluid into the respiratory system, leading to inflammation and infection. Aspiration may be predisposed due to a range of factors such as advanced age sedation, impaired swallowing and sialorrhea [95]. Many of these factors are side effects of clozapine due to its effect on various neuroreceptors [8]. This paper will examine key theories as to the pharmacodynamic effects of clozapine increasing the risk of aspiration. These include (1) anti-cholinergic side effects via muscarinic 1 (M1) and muscarinic 4 (M4) receptors, (2) the effect of histaminergic 1 receptor (H1) action, and (3) extra-pyramidal side effects leading to pneumonia. A visual summary is presented in Figure 8.

Key anti-cholinergic side effects that increase the risk for pneumonia include dry mouth, sedation, reflux of gastric contents, reduction in peristalsis, and confusion [14,27]. Sedation can lower consciousness, lead to a loss of protective reflexes such as the cough reflex and decrease oesophageal tone, which overall makes it easier to aspirate gastric contents [96,97]. This is worsened by decreased gastric mobility [33,34], which leads to gastric contents remaining in the stomach for longer. Dry oral mucosa can thicken the mucosal plug, which is another risk factor for pneumonia [24]. Although dry mouth is a side effect of clozapine use, it is more commonly seen is sialorrhea. Sialorrhea is mainly thought to be driven by muscarinic 3/5 antagonism and muscarinic 4 partial agonism [70]. Sialorrhea is a key contributor to aspiration risk due to the risk of inhalation of saliva into the airways. It is important to notice that clozapine also induces the paradoxical symptoms of sialorrhea which may lead to aspiration [98]. It is found that almost all anticholinergic medications have an increased risk for pneumonia [99,100], though not specifically aspiration. This indicates that anticholinergic action does play an important role in pneumonia risk. However, its importance to aspiration pneumonia specifically needs to be further investigated. As discussed above, clozapine also has a strong affinity for H1 receptors [101]. H1 receptor blockage can lead to the impairment of the laryngeal nerve, leading to poor swallowing and breathing [102,103]. H1 antagonism is also linked with similar side effects to those of anticholinergic activity, such as sedation and drowsiness [104,105].

It is also implicated that clozapine may have anti-dopaminergic effects [106] which can lead to the extrapyramidal side effects of dystonia, tardive dyskinesia, and pseudo-parkinsonism [107]. While these symptoms do increase the risk of pneumonia and are a potential side effect of clozapine [108], clozapine has been shown to cause extra-pyramidal side effects less commonly than many other medications, particularly FGAs [109], which are associated with comparatively lower pneumonia risk [30]. This is further reflected in the weaker affinity of clozapine for dopamine 2/dopamine 3 receptors compared to other antipsychotics, such as risperidone, which had a weaker risk for pneumonia [13]. Hence, it appears that anti-dopaminergic side effects are not the main driver for pneumonia risk in clozapine.

### 4.7. Mortality and Morbidity Due to Clozapine

Pneumonia is a very important yet underappreciated risk factor for clozapine mortality and morbidity [43]. It is the greatest specified cause of death in clozapine users [9] and is associated with more frequent and longer hospitalisations [29], increased intensive care admission and mechanical ventilation [110]. A paper by Han [25] found that pneumonia was the most common reason for hospital admission (32.2%), followed by gastrointestinal causes (19.8%), and then cardiac causes (11.7%). In total, 26% of pneumonia patients needed ICU admission, and 22% needed surgical intervention. Further, clozapine side effects and mortality are cited as key reasons for clozapine cessation. A report [91] found that the quoted reason for stopping clozapine for 35.4% of cases was due to adverse reactions, and that for 13.4% of cases was due to death. These were compared to other anti-psychotics, such as risperidone, which was stopped for the above reasons in 1.9% and 19.9% of cases, respectively. However, clozapine-associated pneumonia is often not listed as an important drug reaction for patients [111]. Both patients and clinicians need to be aware of the particularly elevated risk that pneumonia poses for clozapine users and to actively monitor risk factors.

On the other hand, it is important to acknowledge that, overall, clozapine does not increase mortality or morbidity compared to clozapine or first-generation antipsychotics. A Lancet systematic review [112] found that clozapine is associated with substantially lower mortality compared to other antipsychotics, particularly for suicide. This is supported by multiple papers [113,114,115] that indicate that clozapine has significant benefits to mortality and the control of psychotic symptoms. It is found that clozapine also significantly decreases the overall rates of hospitalisations [7]. They found that clozapine was not associated with a significant increase in cardiovascular risk in all Western countries except Australia, where there is a 10–100-fold greater risk among patients [46]. Other systematic reviews [116,117] found that treatment-refractory schizophrenia untreated with clozapine had double the risk for pneumonia compared to that treated with clozapine and that clozapine significantly reduced the risk of hospitalisation for any reason compared to other SGAs overall [7].

It is proposed that the key underlying risk for the elevated risk for all adverse drug reactions related to clozapine is elevated dosing [57]. It is known that clozapine has a narrow therapeutic range [118], and increased clozapine serum levels can lead to side effects. A study (Suhas 2021) found that most patients who developed ADRs had elevated serum clozapine/norclozapine and that if clozapine levels were adequately managed, symptoms could be well controlled [119]. As previously discussed, when serum clozapine is elevated, patients may be sedated, and signs of pneumonia can be masked. This can make detection [57], and hence later presentations [110], more difficult. It is found that clozapine does not increase the risk of pneumonia mortality relative to the total number of pneumonia presentations (it is the absolute number that is increased), although clozapine does have an overall worse prognosis [110]. This suggests that perhaps the main challenge is the prevention and detection of pneumonia, not the treatment of the condition. Hence, there is a serious need for the formal recognition of the dangers of pneumonia in clozapine patients, and further strategies such as prioritising vaccination among people with severe mental illness are needed for minimisation of the risk due to pneumonia.

### 4.8. Potential for Pneumococcal/Influenza/COVID-19 Vaccination in Clozapine Patients

Due to the increased risk of pneumonia [29], it is important to promote greater awareness of the need for greater pneumonia surveillance and the prioritisation of clozapine users for pneumococcal vaccination [43]. As shown by this review, clozapine users are at risk for pneumonia and may be immunodeficient due to neutropoenia [120,121]. This has led to clozapine becoming a key driver of mortality and morbidity in clozapine patients. Currently, studies have focused more on COVID-19 and influenza [122], and papers discussing the benefits of pneumococcal vaccination are limited. This suggests that there may be an under-appreciation of the risk pneumonia poses.

Further, there is evidence to suggest that clozapine patients have poor uptake of vaccinations [15,123], including pneumococcal vaccination. However, active encouragement by psychiatric providers has been shown in COVID-19 to significantly increase vaccination uptake [124]. This highlights the important role psychiatric physicians can play not only in managing severe mental illness but also in infection prevention. Current Australian pneumococcal vaccination guidelines recommend vaccination for all peoples with risk conditions, First Nations people over age 50 and Non-Indigenous people over the age of 70 [125]. We are proposing formally recognising all severe mental illnesses as a risk condition and recommending vaccination. Clinicians should also have broader awareness, and regular monitoring for pneumonia by clinicians is needed [126].

### 4.9. Statistical Analysis

Our meta-analysis found large heterogeneity, which can be explained firstly by the small number of included papers in the meta-analyses. For the comparison between clozapine and SGAs, nine papers were included, while the comparison of clozapine with FGAs included only four papers. Secondly, as adjusted odds ratios were not reported in the included papers, comparisons between clozapine and other antipsychotic drugs were performed manually by the authors of this paper. As demographic data were not routinely or uniformly reported across the papers, adjustments to the odds ratios were difficult. The large heterogeneity of the papers also underlies why the comparison between clozapine and FGAs did not reach statistical significance.

There were two outliers in the comparison between clozapine and SGAs. An explanation could be due to the paper by Yang et al., 2013 [33] investigating a patient sample with a high rate of upper respiratory tract infections (47.8% in pneumonia cases and 38.0% in the control group), which may explain the higher rate of pneumonia in the case sample. The other outlier is Milano et al. [18], but this can be explained by the high mean Charleson comorbidity index (CCI) of 6 in the included patient population compared to that in other papers included in the analysis, which only had a mean CCI of 1–2.

### 4.10. Strengths

Our review has some key strengths. Firstly, to our knowledge, we are the first review with a meta-analysis to specifically focus on the risk of pneumonia between clozapine and other antipsychotics. Our review also highlights the leading theories surrounding the mechanism for clozapine-induced pneumonia. We also provide a perspective on the key risk factors and provide recommendations to inform clinical practice. Secondly, our review utilises mixed study designs with cohort, case–control and randomised control trials, which enhances our understanding across diverse populations. We also performed subgroup analyses based on generations 1 and 2 of antipsychotics to provide more accurate results for each group. Finally, we utilised a mixed methods approach to our research to more comprehensively understand both the level of risk as well as potential causes for clozapine-associated pneumonia.

### 4.11. Limitations

Despite the investigators’ best efforts, some limitations to this study exist. Firstly, there is large heterogeneity between the included papers. As adjusted odds ratios were not routinely reported, odds ratios were calculated manually from reported raw data. Secondly, demographic data for the clozapine population was inconsistently reported by included papers. This made adjusting for confounding factors such as comorbidities, sex ratio and severity of mental illness difficult. Thirdly, as each paper investigated a different range of antipsychotics, there is a large variance in the calculated odds ratio for each paper. Fourthly, as non-English papers were excluded from this review, this may have led to a bias due to incomplete data. This would affect the generalisability of our results as non-English speaking countries may differ substantially in their demographic, medical and socioeconomic characteristics from those included in this review. Finally, as seen in our funnel plots, indicating potential publication bias, there are smaller studies with non-significant or negative results that impacted our results. This may be caused by the clinical heterogeneity that may have contributed. As discussed in Section 4.7, due to the lack of reporting of demographic and medical characteristics, adjusting the OR was difficult. Overall, this highlights the need for more large-scale controlled trials to be conducted.

## 5. Conclusions

Current research has established the correlation between second-generation antipsychotics and pneumonia and has identified that pneumonia risk is particularly high in clozapine. There is also extensive discussion that factors such as neutropoenia, hyper-salivation and hyper-sedation are key contributing factors to pneumonia risk. However, the literature currently lacks in the understanding of the biological mechanism underlying clozapine-associated pneumonia and the synergistic relationship clozapine has with other comorbidities and medications, particularly other antipsychotics. More studies should also be performed in the outpatient setting, as many database reviews only reported on inpatients and hospital-acquired pneumonia. Successful protection against pneumonia via vaccination in clozapine users is critical to reduce overall mortality and serious adverse events.

## Figures and Tables

**Figure 1 medicina-60-02016-f001:**
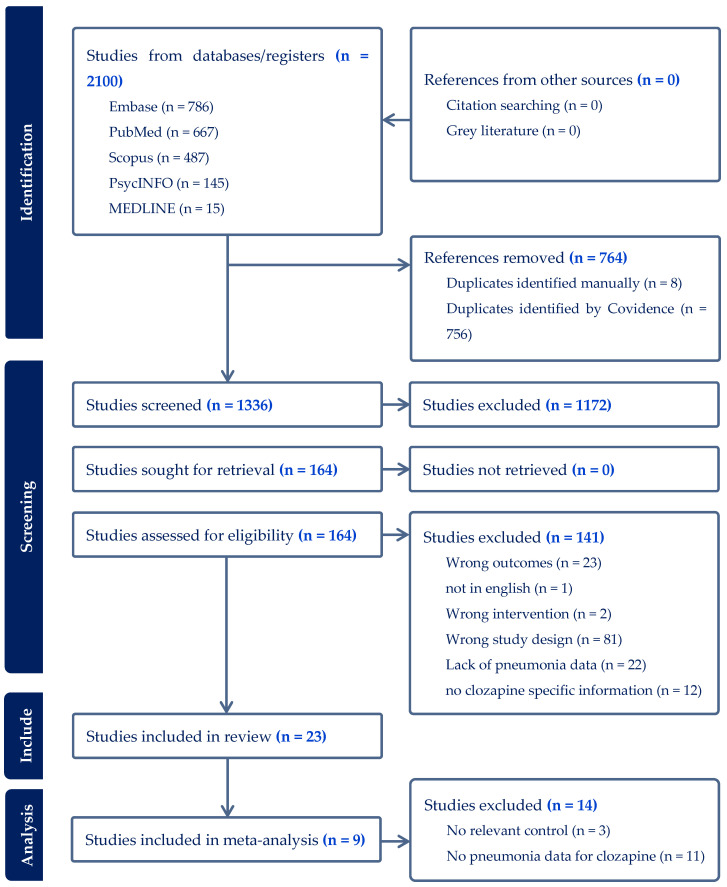
PRISMA 2020 Flowchart Diagram.

**Figure 2 medicina-60-02016-f002:**
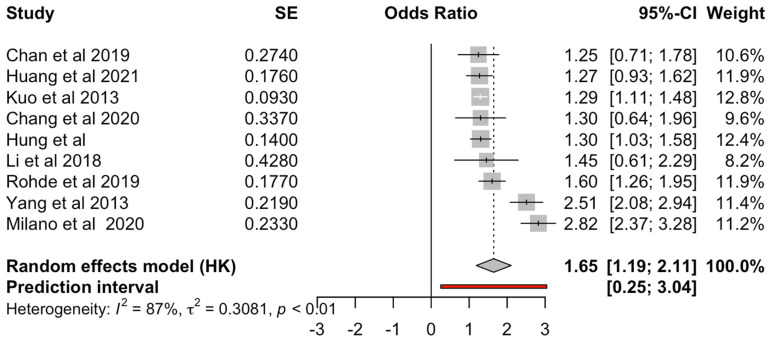
Forest plot for clozapine vs. SGAs [12,14,18,19,20,31,33,34,35].

**Figure 3 medicina-60-02016-f003:**
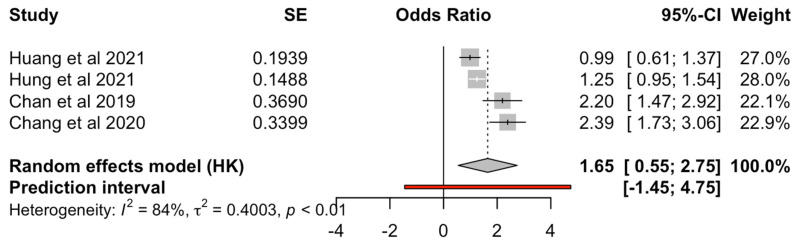
Forest plot for clozapine vs. FGAs [19,20,31,34].

**Figure 4 medicina-60-02016-f004:**
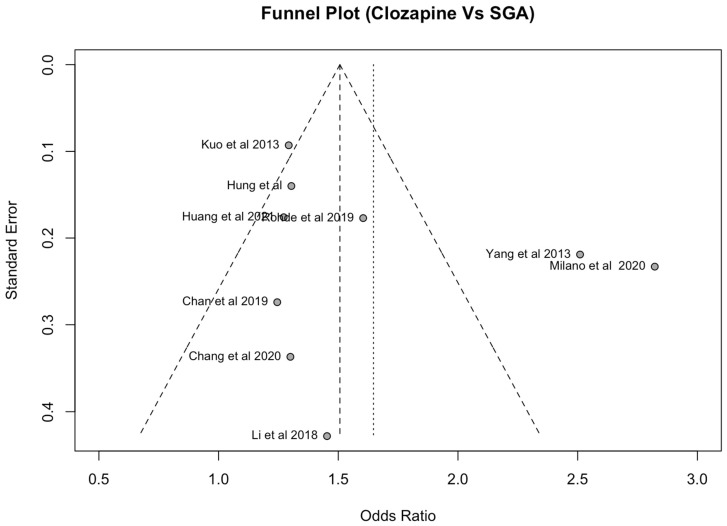
Funnel plot for clozapine vs. SGAs [12,14,18,19,20,31,33,34,35].

**Figure 5 medicina-60-02016-f005:**
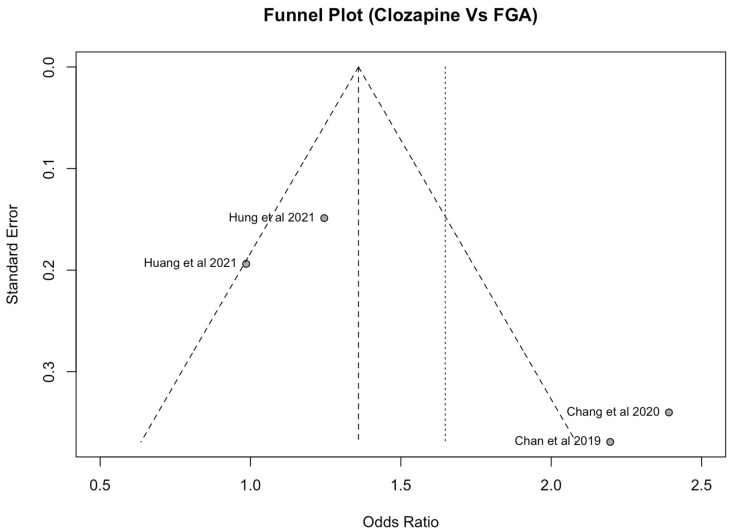
Funnel plot for clozapine vs. FGAs [19,20,31,34].

**Figure 6 medicina-60-02016-f006:**
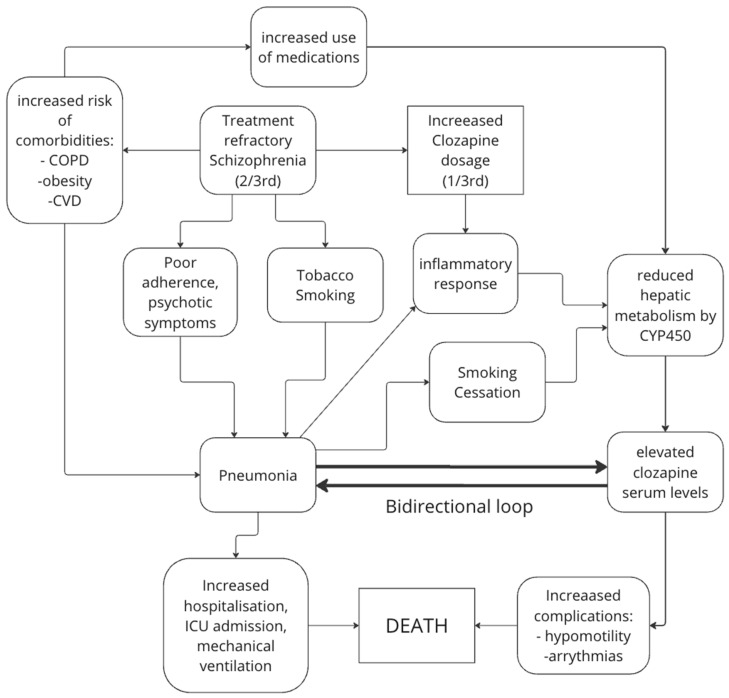
Figure of bidirectional relationship of clozapine and pneumonia.

**Figure 7 medicina-60-02016-f007:**
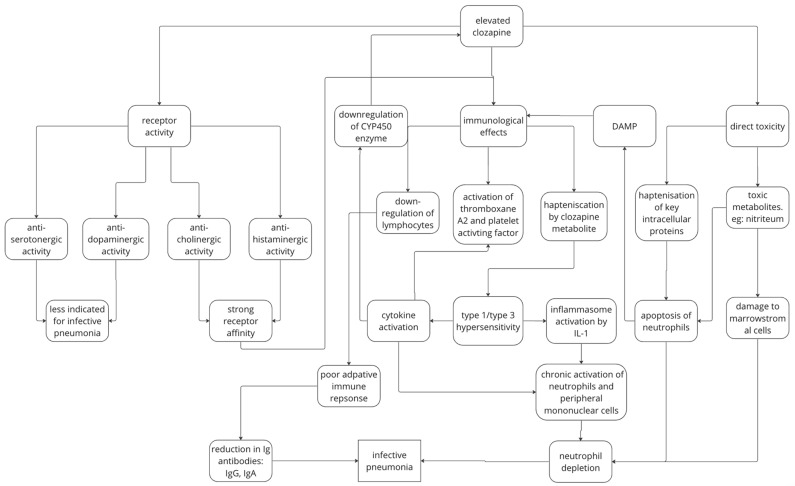
Infection pneumonia mechanism for clozapine.

**Figure 8 medicina-60-02016-f008:**
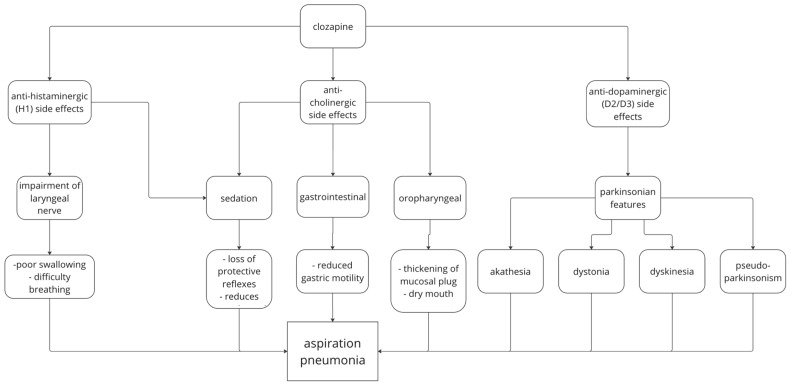
Infection pneumonia mechanism for clozapine use.

**Table 1 medicina-60-02016-t001:** Search terms and related search terms used.

Search Terms	Related Terms
Clozapine	Clozapine, Clopine, Clopine 25, Clozaril, Clozaril 25, Clozitor, Leponex, Zaponex, Denzapine, dibenzodiazepine
Antipsychotics	antipsychotic, antipsychotic, neuroleptic, tranquili*, major tranquili*
Pneumonia	pneumonia, pneumonias, experimental lung inflammation*, lung inflammation*, pneumonitis, pneumotides, pneumocystis, pneumocytes, bacterial pneumonia*, bronchopneumonia*, pneumococcal disease, pneumococcus, lobar pneumonia, Chest infection, acute chest syndrome, acid aspiration syndrome, Mendelson syndrome
Vaccination	Vaccination, vaccine, immunisation, immunization, vaccines
Severe mental illness	Severe mental illness, serious mental illness, enduring mental illness, schizophrenia, schizophrenic, bipolar

Note: * represents the wildcard operator in database search engines.

**Table 2 medicina-60-02016-t002:** Inclusion and exclusion criteria with secondary outcomes screened.

Inclusion criteria: - Papers were accepted within the last 33 years (1 January 1990) in a peer-reviewed journal. - Papers that are primary evidence papers. - Papers are published in English and are full text articles. - Papers that explore at least one of the following: ○ Quantitative data on the rates and/or causes for patients taking clozapine to acquire pneumonia; ○ Quantitative data on patients with treatment-resistant schizophrenia and risks of pneumonia; ○ Quantitative data on patient outcome after developing pneumonia, including mortality. Secondary factors screened for: ○ Demographic risk factors influencing the risk for developing pneumonia in clozapine patients; ○ Comorbidities affecting pneumonia risk and prognosis; ○ Impact of patient diagnosis to pneumonia risk profile; ○ Risk of pneumonia and prognosis for pre-admission compared to post-admission; ○ Potential mechanisms for clozapine-caused pneumonia; ○ Influence on patient mortality due to pneumonia risk and clozapine; ○ Potential for pneumococcal vaccination in patients taking clozapine.
Exclusion criteria - Papers where type of antipsychotics used is not specified or there is no clozapine-specific information; - Papers that do not report original data or are reviews of original data; - Responses to published papers and letter to the editors; - Case reports and collection of case reports; - Papers that discuss pneumonia due to COVID-19; - Papers where participants had active pneumonia prior to initiation of clozapine.

**Table 3 medicina-60-02016-t003:** Quality assessment with Joanna Briggs Institute Checklist Tool for Cohort Studies.

	Question 1	Question 2	Question 3	Question 4	Question 5	Question 6	Question 7	Question 8	Question 9	Question 10	Question 11
Rohde et al., 2018 [17]	Y	Y	Y	Y	Y	Y	Unsure	Y	Y	Y	Y
Milano et al., 2020 [18]	Y	Y	Y	Y	Y	Unsure	Unsure	Y	Y	Y	Y
Chang et al., 2020 [19]	Y	Y	Y	Y	Y	Y	Y	Y	Y	Y	Y
Chan et al., 2019 [20]	Y	Y	Unsure	Y	Y	Unsure	Unsure	Y	N/A	Y	Y
Wu et al., 2019 [21]	Y	Y	Y	Y	Y	N	Y	Y	Y	Y	Y
Yang et al., 2023 [22]	Y	Y	Y	Y	Y	Unsure	Y	Y	Y	Y	Y
DeLeon et al., 2020 [9]	N/A	Y	Y	N	N	Unsure	Y	Y	N/A	Y	N/A ^b^
Rohde et al., 2020 [14]	Y	Y	Y	Y	Y	Y	Y	Y	Y	Y	Y
Haga et al., 2018 [23]	Y	Y	Y	Y	Y	Y	Y	Y	Y	Y	Y
Han et al., 2023 [24]	Y	Y	Y	Y	Y	Y	Unsure	Y	Y	Y	Y
Han et al., 2022 [25]	Y	Y	Y	Y	Y	Y	Unsure	Y	Y	Y	Y
Yang et al., 2021 [26]	Y	Y	Y	Y	Y	Unsure	Unsure	Y	Y	Y	Y
Phaldessai et al., 2019 [27]	Y	Y	Y	N	N	Unsure	Unsure	Y	N	N	N/A ^b^
Factor et al., 2001 [28] ^a^	N/A	Y	Y	Y	N	Y	Unsure	Y	Y	Y	Y
Leung et al., 2017 [29]	Y	Y	Y	Y	Y	N	Unsure	Y	Y	Y	N/A ^b^

^a^ Factor et al., 2001 is an RCT trial by design, but the component used by this study is the cohort study section. ^b^ No statistical analysis was conducted in these papers.

**Table 4 medicina-60-02016-t004:** Quality assessment with Joanna Briggs Institute Checklist Tool for Case-Control Studies.

	Question 1	Question 2	Question 3	Question 4	Question 5	Question 6	Question 7	Question 8	Question 9	Question 10	Question 11
Copeland et al., 2023 [30]	Y	Y	Y	Y	Y	Y	Y	Unsure	Y	Unsure	Y
Kuo et al., 2013 [12]	Y	Y	Y	Y	Y	Y	Y	Y	Y	N	Y
Huang et al., 2021 [31]	Y	Y	Y	Y	Y	Y	Y	Y	Y	N	Y
Stoecker et al., 2017 [32]	Y	Y	Y	Y	Y	Y	Y	Y	Y	Y	Y
Yang et al., 2013 [33]	Y	Y	Y	Y	Y	Y	Y	Y	Y	N	Y
Hung et al., 2016 [34]	Y	Y	Y	Y	Y	Y	Y	Y	Y	Y	Y
Cepaityte et al., 2021 [13]	Y	Y	Y	Y	Y	Y	Y	Y	Y	N	Y
Li et al., 2018 [35]	Y	Y	Y	Y	Y	Y	Y	Y	Y	Y	Y

**Table 5 medicina-60-02016-t005:** Characteristics of 23 included studies with 330,699 participants.

Author	Number of Patients Taking Clozapine	Aim of Papers	Location of Study	Study Format	Length of Time of Study (Years)	Comparison of Rate of Pneumonia Compared to Other Antipsychotics	Discusses Mechanisms for Clozapine-Induced Pneumonia
Han et al., 2022 [25]	120	To explore the characteristics of inpatients with mental disorders who have CAP and analyse the risk factors.	China	retrospective cohort study in medical centre	4	NO	YES
Han et al., 2023 [24]	1417	To analyse the epidemiological characteristics of patients with mental disorders who developed HAP while hospitalised, and to identify the risk factors for HAP in this patient population.	China	retrospective cohort study in medical centre	4	NO	YES
Yang et al., 2021 [26]	N/A	To elucidate the risk factors for HAP in the middle-aged and elderly hospitalised patients with schizophrenia.	China	cohort study in medical centre	4	YES (OR)	YES
Yang et al., 2023 [22]	986	To explore the risk factors for lower respiratory tract infection (LRI), particularly pneumonia.	China	retrospective cohort study in medical centre	4	NO	NO
Rohde et al., 2018 [17]	3262	To investigate rates of myocarditis, pericarditis, and cardiomyopathy in previously clozapine-naïve out-patients initiating clozapine treatment.	Denmark	registry review cohort study (DCRR ^a^, DNPreR ^c^)	19	NO	NO
Rohde et al., 2020 [14]	1872	To explore whether antipsychotics generally increase pneumonia risk, and which antipsychotic are particularly associated. This study also aims to examine whether pneumonia risk for clozapine is greater than other antipsychotics.	Denmark	registry review cohort study (DCRR ^a^, DPCRR ^b^, DNPreR ^c^, DNPat ^d^, DRCD ^e^)	50	YES (percent and incidence rate)	YES
Haga et al., 2018 [23]	34	To elucidates the risk factors for pneumonia in patients with schizophrenia.	Japan	retrospective chart review cohort study	2	YES	YES
Phaldessai et al., 2019 [27]	40	To quantify the indications, titration rates, doses of clozapine, side effects, and outcome and comparing them with existing data.	New Zealand	health records review cohort study (WDHD ^i^)	10	NO	NO
Kuo et al., 2013 [12]	8688	To explore various dimensions of the associations between each second-generation antipsychotic drug and the risk of pneumonia.	Taiwan	database review case–control (NHIRD ^f^)	16	YES (OR + absolute value)	YES
Yang et al., 2013 [33]	96	To explore the association between antipsychotic and mood stabilisers and the risk of pneumonia to provide evidence for clinical practice.	Taiwan	database review case–control (NHIRD ^f^)	11	YES (OR)	YES
Hung et al., 2016 [34]	323	To investigate the risk of pneumonia associated with the use of antipsychotic drugs in older-adult patients with Parkinson’s disease (PD) in Taiwan.	Taiwan	case–control study	24	YES (OR + absolute value)	YES
Li et al., 2018 [35]	11	To investigate the effect of antipsychotic use on the risk of URI progression to pneumonia in patients with schizophrenia.	Taiwan	registry review cohort study (NHIRD ^f^)	20	YES (hazard ratio)	YES
Chan et al., 2019 [20]	186	A comprehensive analysis of factors potentially associated with risk of pneumonia in psychiatric inpatients.	Taiwan	cohort study in medical centre	6	YES (absolute numbers)	NO
Wu et al., 2019 [21]	22,774	To investigate the risk factors of pneumonia in patients with schizophrenia who use clozapine.	Taiwan	database review cohort study (NHIRD ^f^)	14	NO	YES
Chang et al., 2020 [19]	809	To explore the recurrent pneumonia risk after re-exposure to antipsychotics	Taiwan	Nested Case–control registry review (NHIRD ^f^)	10	YES (OR + total number)	NO
Huang et al., 2021 [31]	780	To explore the possibility of psychiatric medications (including antipsychotics) increasing the incidence of recurrent pneumonia.	Taiwan	database review cohort study (NHIRD ^f^)	16	YES- as risk ratios	YES
Copeland et al., 2023 [30]	29,836	To examine the association of pneumonia-related death with specific antipsychotic exposure.	United Kingdom	case–control study	23	YES (OR)	YES
Factor et al., 2001 [28]	53	To report the results of the PSYCLOPS in the treatment of Parkinsonism trial which examined the chronic safety and efficacy of clozapine in the treatment of drug-induced psychosis in Parkinson’s disease (PD).	United States	randomised control trial (PSYCLOPS ^j^ trial)	12 weeks	NO	YES
Leung et al., 2017 [29]	104	To describe the primary reasons why clozapine users were admitted to a nonpsychiatric medical unit.	United States	single-centre, retrospective cohort study, chart review	12	NO	YES
Stoecker et al., 2017 [36]	155	To determine whether the incidence of pneumonia in patients taking clozapine was more frequent compared with those taking risperidone or no atypical antipsychotics.	United States	retrospective case–control study in medical centre	2	YES (OR + absolute value)	NO
Milano et al., 2020 [18]	134	To compare the rate of hospitalisations for pneumonia in patients with a psychotic or bipolar disorder who were prescribed 1 of 4 second-generation antipsychotics prior to admission.	United States	cohort study in medical centre	2	YES (OR)	YES
Cepaityte et al., 2021 [13]	119,019	To enhance the current understanding of individual antipsychotics safety by using a previously published methodology with a combined pharmacovigilance-pharmacodynamic approach.	United States	case–control database review (FAERS ^g^)	15	YES (OR)	NO
DeLeon et al., 2020 [9]	>140,000	To explore the role of pneumonia in clozapine-associated adverse events.	Multiple countries	database review cohort study (Vigibase data ^h^)	51	YES (percentage)	YES

^a^ DCRR—Danish Civil Registration Register. ^b^ DPCR—Danish Psychiatric Central Research Register. ^c^ DNPrER—Danish National Prescription Register. ^d^ DNPatR—Danish National Patient Register. ^e^ DCRD—Danish Civil Record of Death. ^f^ NHRID—National Health Insurance Research Database. Based in Taiwan. ^g^ FAERS—FDA Adverse Event Reporting System. ^h^ Vigibase Data—This refers to the 2020 World Health Organisation Vigibase Data for Adverse Drug Reactions. ^i^ WDHD—Danish National Patient Register. ^j^ PSYCLOPS- Psychosis and Clozapine in Parkinson’s Disease Study.

**Table 6 medicina-60-02016-t006:** Demographic information of Participants from Included studies.

Author	Number of Cloapine Users	Gender (F %)	Mean Age (Years)	Comorbidities—Charlson Comorbidity Index (CCI)	Diagnosis
Demographic Characteristics of papers with Clozapine user specific information					
Rohde et al., 2020 [14]	1872	43%	38.5	0 comorbidity = 24.9%, 1 comorbidity = 64%, ≥2 comorbidities = 11.1%	Schizophrenia
Hung et al., 2016 [34]	323	34.8%	49.6	0–1 comorbidity = 76.8%, 2 comorbidities = 17.3%, 2+ comorbidities = 6%	Schizophrenia
Stoecker et al., 2017 [32]	155	36.8%	57.2	% for each comorbidity given	Schizophrenia, Schizoaffective disorder, bipolar disorder
Wu et al., 2019 [21]	4278 ^a^	43.3%	N/A	0 comorbidity = 68.4%, 1 comorbidity = 20.4%, 2+ comorbidities = 11.2%	Schizophrenia
DeLeon et al., 2020 [9]	>140,000	N/A	N/A	N/A	N/A
Leung et al., 2017 [29]	104	33.0%	52	% of disease prevalence given	Schizophrenia
Phaldessai et al., 2019 [27]	40	57.5%	74.8	N/A	Parkinson’s disease
Demographic information for papers without information on clozapine users. Information relates to whole population.					
Milano et al., 2020 [18]	134	44.20%	58.5	Mean CCI-6 comorbidities	Schizophrenia, schizoaffective disorder, bipolar disorder
Chang et al., 2020 [19]	809	N/A	N/A	details of specific comorbidities provided	Schizophrenia
Cepaityte et al., 2021 [13]	14,401	45%	54.1	N/A	Not reported
Copeland et al., 2023 [30]	29,836	28%	43.2	N/A	Not reported
Kuo et al., 2013 ^b^ [12]	8688	37%	42.8	1 comorbidity = 78.3% 2 comorbidities = 16.5% 2+ comorbidities = 5.3%	Schizophrenia
Kuo et al., 2013 ^c^ [12]	8688	37%	42.8	1 comorbidity = 83.7% 2 comorbidities = 13.6% 2+ comorbidities = 2.7%	Schizophrenia
Huang et al., 2021 [31]	780	53%	77	Charlson comorbidity scores given in numbers	Parkinson’s disease
Li et al., 2018 [35]	11	N/A	N/A	0–1 Comorbidity-73.9, 2 comorbidities-15.4% 3 comorbidities-10.6%	Bipolar disorder
Yang et al., 2013 [33]	96	38.9%	44.2	0 comorbidity-43.1% 1 comorbidity-33.6% 2+ comorbidities-13.7%, 3+ comorbidites-0.6%	Bipolar disorder
Chan et al., 2019 [20]	194	54%	47.6	details of specific comorbidities provided	Schizophrenia
Yang et al., 2023 [22]	986	58%	44.1	N/A	Schizophrenia, depression
Rohde et al., 2018 [17]	3262	43%	46.3	0 comorbidity-22.65% 1 comorbidity-58.15% ≥2 comorbidities = 19.19%	Not reported
Haga et al., 2018 [23]	34	N/A	N/A	Details of specific comorbidities provided	Schizophrenia
Han et al., 2023 [24]	1417	57%	31	0–1 comorbidity (86.46%) 2 comorbidities-(7.01%) ≥3 comorbidities-(6.53%)	Schizophrenia
Han et al., 2022 [25]	120	46%	50	percentages of specific comorbidities provided	Not reported
Yang et al., 2021 [26]	N/A ^d^	65%	N/A	details of specific comorbidities provided	Schizophrenia
Factor et al., 2001 [28]	53	N/A	N/A	N/A	Parkinson’s disease

Note: ^a^ test cohort. ^b^ Charlson comorbidity index for patients who developed pneumonia. ^c^ Charlson comorbidity index for patients who developed pneumonia. ^d^ There is no detailed clozapine participant number in this study, but the total number of participants is 2617.

**Table 7 medicina-60-02016-t007:** Information regarding prevalence and risk of pneumonia.

Year of Publication and Author	Reporting Criteria for Pneumonia	Risk of Pneumonia Based on Clozapine Use ^a^ (Odds Ratio)	Odds Ratio Reference	Prevalence of Clozapine Induced Pneumonia	Cause of Pneumonia (asp ^h^, HAP ^i^, CAP ^j^, inf ^k^)
Rohde et al., 2020 [14]	ICD-10 ^b^: J12–19	1.54	other antipsychotics	1.87% ^l^	N/A
Milano et al., 2020 [18]	Not reported	2.37	other antipsychotics	N/A	N/A
Chang et al., 2020 [19]	ICD-9 ^b^ codes 480–486 or 507	1.95	other antipsychotics	N/A	N/A
Hung et al., 2016 [34]	ICD-9 ^b^ codes 480–486 or 507	1.47 ^c^	termination of clozapine and no reuse	29.90% ^n^	N/A
Hung et al., 2016 [34]	ICD-9 ^b^ codes 480–486 or 507	2.02 ^d^	termination of clozapine and no reuse	29.90% ^n^	N/A
Hung et al., 2016 [34]	ICD-9 ^b^ codes 480–486 or 507	1.99 ^e^	termination of clozapine and no reuse	29.90% ^n^	N/A
Cepaityte et al., 2021 [13]	Not reported	4.8 ^f^	Haloperidol	N/A	asp, inf
Cepaityte et al., 2021 [13]	Not reported	2.1 ^g^	Haloperidol	N/A	asp, inf
Copeland et al., 2023 [30]	Post-mortem reporting	N/A	N/A	N/A	Asp, HAP
Kuo et al., 2013 [12]	ICD-9 ^b^ codes 480–486 or 507	1.47 ^c^	other antipsychotics	25.2% ^c^	N/A
Kuo et al., 2013 [12]	ICD-9 ^b^ codes 480–486 or 507	0.93 ^d^	other antipsychotics	22.7% ^d^	N/A
Kuo et al., 2013 [12]	ICD-9 ^b^ codes 480–486 or 507	3.18 ^e^	other antipsychotics	39.7% ^e^	N/A
Huang et al., 2021 [31]	Not reported	1.08 ^c^	other antipsychotics	29.86%	N/A
Huang et al., 2021 [31]	Not reported	1.41 ^d^	other antipsychotics	29.86%	N/A
Huang et al., 2021 [31]	Not reported	1.77 ^e^	other antipsychotics	29.86%	N/A
Stoecker et al., 2017 [32]	ICD-9 ^b^ codes, clinical diagnosis	4.07 ^m^	patients not on antipsychotics	34%	Asp, HAP, CAP
Li et al., 2018 [35]	ICD-9 ^b^ codes 480–486 and 507	1.25 ^e^	clozapine initial use	N/A	CAP
Yang et al., 2013 [33]	ICD-9 ^b^ codes 480–486 and 507	3.73	patients not on antipsychotics	N/A	N/A
Chan et al., 2019 [20]	CXR or chest CT with clinical parameters	1.55	Not reported	N/A	N/A
Wu et al., 2019 [21]	ICD-9 ^b^ codes 480–486 and 507	N/A	N/A	2.98% ^o^	N/A
Yang et al., 2023 [22]	CXR or chest CT with clinical parameters	2.18	other antipsychotics	N/A	N/A
DeLeon et al., 2020 [9]	Not reported	N/A	N/A	4.99%	N/A
Rohde et al., 2018 [17]	Not reported	N/A	N/A	N/A	inf
Leung et al., 2017 [29]	Not reported	N/A	N/A	18.50%	N/A
Haga et al., 2018 [23]	CXR or chest CT with clinical parameters	N/A	N/A	2.90%	N/A
Han et al., 2023 [24]	CXR or chest CT with clinical parameters	3.93	CC1 0–1	N/A	HAP
Han et al., 2022 [25]	CXR or chest CT with clinical parameters	3.21	Patients not on antipsychotics	N/A	CAP
Yang et al., 2021 [26]	Not reported	1.81	Other antipsychotics	N/A	HAP
Phaldessai et al., 2019 [27]	Not reported	N/A	N/A	5%	N/A
Factor et al., 2001 [28]	Not reported	N/A	N/A	13.20%	N/A

^a^ Describes current clozapine use unless otherwise specified. ^b^ ICD = International Classification of Diseases. ^c^ Describes the odds ratio of patients who have never been on clozapine before or after the initial pneumonia event. ^d^ Describes the odds ratio of patients who were started on clozapine only after the initial pneumonia event. ^e^ Describes the odds ratio of patients who were on clozapine before the initial pneumonia event and then recommenced on clozapine again. ^f^ Describes infective pneumonia odds ratio. ^g^ Describes the aspiration odds ratio. ^h^ Aspiration pneumonia. ^i^ Hospital-acquired pneumonia. ^j^ Community-acquired pneumonia. ^k^ Infective pneumonia (may include HAP and CAP). ^l^ The incidence is 2.14/100-person years. ^m^ HAP odds ratio = 14.10, CAP odds ratio = 3.87, aspiration pneumonia odds ratio = 3.70. All comparisons were made to the general population; 7.7/100 person-years is the incidence of recurrent pneumonia, 1.12/100 person-years is the incidence of new pneumonia. ^n^ Prevalence of recurrent pneumonia among new use and re-exposure groups combined. ^o^ The incidence rate of pneumonia among patients using clozapine.

## Data Availability

No new data were created or analyzed in this study. Data sharing is not applicable to this article.

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
