# Peer review of "Clozapine and Pneumonia: Synthesizing the Link by Reviewing Existing Reports—A Systematic Review and Meta-Analysis"

_medicina, 2024, doi:10.3390/medicina60122016_

Round 1

Reviewer 1 Report

Comments and Suggestions for Authors

Peer review report: Manuscript on Clozapine and Pneumonia Risk.

Dear authors  Victor Zhao, Yiting Gong, Naveen Thomas, and Soumitra Das,

    Thank you for allowing me to review your manuscript: "Clozapine and Pneumonia: Identifying the Connection Using The Published Literature — A Systematic Review and Meta-Analysis". It was an honor to read you thoroughly discussing such a critical and complex area of clinical psychiatry. And I want to note that there was a lot of work and careful analysis behind this manuscript. Your commitment to providing the light on clozapine risks, especially pneumonia, offers an important voice in the debate over how to best manage antipsychotics in at-risk patients.
       Your systematic review and meta-analysis provides a timely and welcome summary of the available evidence, and is needed to clear up a space where reports conflict. Your adoption of various study designs (RCTs, cohort, and case-control studies) and PRISMA guidelines are two strengths of your approach and demonstrate your commitment to strict criteria. Moreover, your focus on mechanisms (clozapine-induced hypersedation, sialorrhoea, immunomodulatory responses in neutrophils) offers an additional perspective that will add to our ability to better understand why people taking clozapine are more likely to develop pneumonia.
      For now, there is no coherent link between biological process and clinical effects in research on clozapine side-effects. What you are doing, not only investigating the relationship to pneumonia, but also recommending practical preventive interventions, like pneumococcal vaccine, is a great gap. This proactive approach lands your manuscript in a prime position to shape clinical practice and highlights its relevance to safer antipsychotic prescribing.
       I also want to thank you for looking into the double-edged sword of clozapine use and infection risk. It’s an intricate account of how changes in clozapine serum levels over time during infection can lead to a harmful feedback loop, which is one area where less has been written. If you are able to illuminate these intricate dynamics, your manuscript opens the possibility of further study on individual monitoring for clozapine users, which is a promising future direction.
   In the paper "Clozapine and Pneumonia: Synthesizing the Link by Reviewing the Existing Reports — A Systematic Review and Meta-Analysis," you discuss the relationship between clozapine and increased risk of pneumonia. It’s to provide a clear picture of demographic, health, and pharmacological drivers of this risk. They use a systematic review procedure, using the PRISMA guidelines, and provide a meta-analysis.

     The topic is timely, given the prevalence of pneumonia in people who take clozapine, and with the ongoing effort to maximise the efficacy of antipsychotic therapy in serious mental illness.

Detailed evaluation

A) Objectives and rationale

In the paper, its purpose is clear, that is to investigate the association between clozapine and the risk of pneumonia (Page 1, Lines 9–15). Yet the reasons why this particular study might be warranted can be framed in relation to gaps in the literature.

Recommendation 1: Point out explicitly that this review is new, with recent contradictory results about how clozapine-induced pneumonia works (Page 2, Lines 45–55).

Recommendation 2: Stress the need to address the nefarious underuse of pneumococcal vaccine among clozapine users as a preventive measure (Page 1, Lines 25–26).

B) Replicability and methodological clarity

The approach is PRISMA compliant, and PubMed, PsycINFO, and MEDLINE are all relevant databases (Page 2, Lines 63–70).
Proposal 1: Include more information about how the screen was conducted using COVIDENCE and how disputes between reviewers were settled (Page 2, Line 69).

Suggestion 2: Add details about the data extraction steps, e.g. what steps were taken to keep the studies consistent (Page 3, Lines 75–78).

C) Statistical analysis

Your statistical approach is mostly good, based on random effects and sensitivity analysis (Page 3, Lines 79–91). However, certain aspects need clarification.
Recommendation 1: Explain why there was no subgroup analysis even though demographic factors were mentioned as driving heterogeneity (Page 3, Lines 83–85).
Option 2: Relying on unweighted odds ratios can lead to bias. Consider doing a meta-regression if possible for the confounding variable (Page 4, Lines 86–90).

D) Tables and Figures

The tables and figures are extensive, though they could be cut down to some useful points.
Tip 1: Merge Tables 6 and 7 to get one comprehensive picture of study-by-study demographic data (Page 4, Lines 116–167).

Recommendation 2: Provide confidence intervals in Figures 2 and 3 to express how accurate are estimates of effects (Page 4, Lines 231–235).

E) Interpretation of Results

Results can be best interpreted using the data (Page 6, Lines 273–287). But there are other fields in which the discussion could be widen.
Recommendation 1: The comment about the inverse relationship between clozapine and infection risk requires more discussion, especially with regard to immune modulation (Page 10, Lines 477–487).
Suggestion 2: Deal with the potential confusion caused by quitting smoking regarding clozapine and the incidence of pneumonia (Page 10, Lines 357–360).

F) Strengths of the Study

The manuscript emphasizes its power as the first full meta-analysis in this area (Page 10, Lines 287–290).

Comment: Promote mixed study designs (RCTs, cohorts, case-control studies) as a strength so that you can get a good feel for the impact of clozapine on various populations (Page 5, Lines 124–131).

G)Limitations

The limitations are admitted (Page 12, Lines 78–85), but they deserve more.
Tip 1: Say it might be biased because the studies were excluded from non-English study, and that would prevent generalisability (Page 2, Lines 66–67).
Suggestion 2: Comment on the publication bias reflected by the funnel plot asymmetry (Page 12, Lines 262–267).

H) Structure and flow

It’s a nicely formatted manuscript, though one that could use some slight tweaking.
Change proposal 1: Divide Section 4.5 Mechanisms of Pneumonia into clearer sections for Infective and Aspiration Pneumonia (Page 11, Lines 425–490).
Tip 2: Introduction can be cut short to avoid repeating the same things that have been discussed in the abstract (Page 1, Lines 30–40).
I) Language and grammar

The language is usually formal and professional, with a few cases of elaborate sentence structures that could be reduced.
Recommendation: A comprehensive editing of the words should be done to make them easier to understand and to reduce the jargon (e.g., "haptenization-led apoptosis" can be rendered into a more palatable or short explanation; Page 12, Line 473).

I hope that you will carry forward this worthy line of enquiry. Antipsychotic pharmacotherapy is an ever-changing field, and your work provides insights that might be used to inform future clinical practice. This is a rich field for extension, for example, to examine actual patient outcomes in the outpatient clinic or to examine how concomitant medication and lifestyle combine to reduce pneumonia risk in clozapine users.

You have been passionate about advancing our knowledge about clozapine risks from beginning to end, which you can see throughout the manuscript. It has been an honor to read and work with you and look forward to your contributions to the field. I’m again grateful for your input to this important field of study.

Yours truly

Serving Peer Reviewer at Medicina MDPI

Comments on the Quality of English Language

On the whole, English is well written and does not hinder the reading of the manuscript. There are places, however, where there is room for improvement. The writing is complicated with sentence constructions and sometimes specialized terminology not easily accessible to an unread audience. I recommend the following:  

Compound sentences, primarily in the paragraphs on biological processes (i.e., Page 11, Lines 425–490). That will be more readable without altering the science. 

Try to avoid using a lot of scientific terminology, or try to describe it as concisely as possible for the things you don’t know how to say (like "haptenization-mediated apoptosis" on Page 12, Line 473). 

Apply a general language edit for grammatical mistakes, especially in the introduction and discussion sections, to get everything flowing. 

Author Response

Dear Reviewer,

Thank you kindly for your feedback and comments regarding our paper. We greatly appreciate the time and energy in which you have spent reviewing our paper and helping us improve our work. We are particularly grateful for your extensive and specific recommendations. After careful consideration, we have implemented many of your recommendations, which we believe have greatly improved our paper. We have highlighted our changes in red. If there is anything else we can do, please do not hesitate to let us know. Please see below for our responses:

Point by Point Response:

Recommendation 1: Point out explicitly that this review is new, with recent

contradictory results about how clozapine-induced pneumonia works (Page 2,

Lines 45–55).

Thank you for your suggestions. A key outcome of our research was to clarify the conflict around the causes and mechanisms that cause pneumonia in clozapine users. We have edited our paper to explicitly mention that we are a new review specific to both clozapine and pneumonia. We have also highlighted the gap our research fills more clearly. Please find our changes on Page 2, Lines 66-71.

Recommendation 2: Stress the need to address the nefarious underuse of

pneumococcal vaccine among clozapine users as a preventive measure (Page 1,

Lines 25–26).

Thank you for your suggestion. Vaccination is a key recommendation we would like to make, so we agree that it should be stressed in our introduction. We have edited our paper to highlight this point. Page 2, Lines 72-75.

Proposal 1: Include more information about how the screen was conducted

using COVIDENCE and how disputes between reviewers were settled (Page 2,

Line 69).

Thank you for your feedback. We have incorporated your feedback into our methods and described in more detail how COVIDENCE was used and how disputes were resolved. Page 2, Lines 91-94.

Suggestion 2: Add details about the data extraction steps, e.g. what steps were

taken to keep the studies consistent (Page 3, Lines 75–78).

Thank you for your feedback. We agree that our details on data extraction were too brief, and more details are needed. We have included more details of our data extraction process. Page 3, Lines 120-130.

Recommendation 1: Explain why there was no subgroup analysis even though

demographic factors were mentioned as driving heterogeneity (Page 3, Lines

83–85).

We observed significant heterogeneity in our analysis and conducted subgroup analyses for first-generation (Figure 3) and second-generation (Figure 2) studies. However, despite these subgroup analyses, the limited number of included papers contributes to the heterogeneity.

Option 2: Relying on unweighted odds ratios can lead to bias. Consider doing a

meta-regression, if possible, for the confounding variable (Page 4, Lines 86–90).

Thank you for your recommendation. We included nine papers in our analysis; however, meta-regression is generally not recommended when fewer than ten studies are available. We have explained this in the methodology section. Also, we have weighted our odds ratio. Please have a look at Figure 2 and Figure 3.

Tip 1: Merge tables 6 and 7 to get one comprehensive picture of study-by-study

demographic data (Page 4, Lines 116–167).

Thank you for your feedback. We have merged Tables 6 and 7 as per your recommendation. Page 15

Recommendation 2: Provide confidence intervals in Figures 2 and 3 to express

how accurate are estimates of effects (Page 4, Lines 231–235).

Thank you for your recommendation. We have provided confidence intervals visually and in numeric form in Figures 2 and 3 on Page 25 and Page 26.

Recommendation 1: The comment about the inverse relationship between

clozapine and infection risk require more discussion, especially about

immune modulation (Page 10, Lines 477–487).

Thank you for your comment regarding the section ‘4.3 Acute and chronic clozapine use’. We have expanded our discussion on the correlation between the length of clozapine use and the risk of pneumonia. These changes can be seen on Page 30, Lines 518-523. We have also discussed the potential mechanisms of immune modulation in Figure 8 and Section 4.6.1 on pages 32-33, Lines 627- 652.

Suggestion 2: Deal with the potential confusion caused by quitting smoking

regarding clozapine and the incidence of pneumonia (Page 10, Lines 357–360).

Thank you for your feedback. We have edited out phrasing to resolve any confusion. Page 32, Lines 510 to 515.

Comment: Promote mixed study designs (RCTs, cohorts, case-control studies)

as a strength so that you can get a good feel for the impact of clozapine on

various populations (Page 5, Lines 124–131).

Thank you for your suggestions regarding the strengths of our paper. We agree that it is important to have a section dedicated to discussing the strengths of our research to inform viewers. We have taken on board the feedback you have kindly provided and added it to our review. Please find the addition on Page 41, Lines 99-109.

Tip 1: Say it might be biased because the studies were excluded from non-

English study, and that would prevent generalisability (Page 2, Lines 66–67).

Thank you for your feedback, please find our addition on Page 42, Lines 129-132.

Suggestion 2: Comment on the publication bias reflected by the funnel plot

asymmetry (Page 12, Lines 262–267).

Thank you for your feedback. Publication bias is an important limitation of our research, and we agree to mention it in our limitations. Additionally, the small number of included studies and the limited sample size of participants further contribute to this limitation. Page 42, Lines 133-137.

Change proposal 1: Divide Section 4.5 Mechanisms of Pneumonia into clearer

sections for Infective and Aspiration Pneumonia (Page 11, Lines 425–490).

Thank you for your recommendation, we have divided the mechanisms into infective and aspiration pneumonia 4.6.1 and 4.6.2. Page 33-35, Lines 749-849 and Page 38, Lines 2-39.

Tip 2: Introduction can be cut short to avoid repeating the same things that have

been discussed in the abstract (Page 1, Lines 30–40).

Thank you for your recommendation. We have tried to add in the additional features recommended by reviewers and have removed repetition between our abstract and introduction. Please see our edited introduction on Page 1, Lines 30-75.

Recommendation: A comprehensive editing of the words should be done to

make them easier to understand and to reduce the jargon (e.g., "haptenization-

led apoptosis" can be rendered into a more palatable or short explanation: Page

12, Line 473).

Thank you for this feedback. We agree that readability is important, so we have reviewed our paper for difficult-to-understand terms such as hapten and marrowstromal cells and explained them accordingly. Our explanation for a hapten can be found on page 34, Lines 794-796. Two reviewers have done a manual review of the article to ensure no spelling and grammar errors. We have also used spell-check and grammar software to check for further errors.

Reviewer 2 Report

Comments and Suggestions for Authors

Thanks for the good and useful article.

Comments:

Article idea: It is good. attention has been given to both the  drug side effects and the need for it( especially persistent psychosis) and suggestions for managing the side effects.

Title: acceptable 

Abstract: acceptable

Introduction: About the beneficial effects of clozapine, especially in treatment-resistant patients, more explanation should be given so that it dose not cause not be use it, and the harms and benefits can be seen side by side.

Methods: acceptable. but even so, it is better to bring more references from the articles that suggested the vaccine for prevention.

Results: acceptable 

Discussion: There should be more discussion about the cause of pneumonia and whether it is primary or secondary, as well as the relationship with myocarditis and leukopenia or others.

Conclusion: acceptable

Reference: acceptable

Author Response

Dear Reviewer,

Thank you for your time and efforts in reviewing our paper. We have carefully reviewed your feedback and taken on-board your feedback. If there is anything else, please do not hesitate to let us know how to improve. Please find attached our responses or see below:

Comment 1: Introduction: About the beneficial effects of clozapine, especially in treatment-resistant patients, more explanation should be given so that it does not cause not to be used, and the harms and benefits can be seen side by side.

Thank you for this important suggestion. We agree that despite the risks and unfavourable side effects profile, clozapine remains an important medication for the treatment of treatment-refractory schizophrenia due to its clinical efficacy in controlling positive side effects and reducing overall mortality and hospitalization. We have made an addition in our introduction reflecting this. Page 1, Lines 36-40.

Comment 2: It is better to bring more references from the articles that suggested the vaccine for prevention.

Thank you for your comments regarding our methodology. We agree that we should explain more about how we performed our search for vaccination. Please find our explanation on Page 2, Lines 86-87. Additional information may also be found in Table 1 and Supplementary Appendix B.

Comment 3: Discussion: There should be more discussion about the cause of pneumonia and whether it is primary or secondary, as well as the relationship between myocarditis and leukopenia or others.

Thank you for your comment. We agree that we can expand on our discussion more and have added two sections that discuss the causes of pneumonia for both infective and aspiration pneumonia. Page 33, Lines 750-753 and Page 37, Lines 2-7. In terms of the link between clozapine and leukopenia, we have explored the link between clozapine and neutropenia/agranulocytosis, which increases the risk for pneumonia on Page 34, Lines 791- 816. We have also explored the link between clozapine and pneumonia through extra-pyramidal symptoms and anti-cholinergic side effects in Figure 6 and section 4.6.2 on Page 38.

We have added to our discussion regarding the link between clozapine-associated pneumonia and other adverse events, such as myocarditis and leukopenia. Page 34, Lines 731-733.

Reviewer 3 Report

Comments and Suggestions for Authors

General comments:

This is a very interesting review of an atopic antipisotic (clozapine) widely used in clinical practice.

Good methodology. The results are well presented with excellent discussion and a very extensive bibliography. Congratulations for the great work done.

Specific comments:

.- Introduction. In the first sentence the term "schizophrenia" is repeated twice. Please, review.

.- Tables. Table 5. Characteristics of Included Studies. Perhaps the studies could be ordered by country and by year of publication.

Perhaps the number of studies and patients could be included in the title. Also consider adding a column that includes the number of patients from each of the 23 studies.

Maybe tables 5-7 could be merged.

.- Discussion. Strengths section is missing.

.- References. 112 quotes are included. 52 (46.4%) of which are recents, that is, 5 years or less from their publication. Congrats!

References 100 and 111. Are they completed?

Author Response

Dear Reviewer,

Thank you for taking the time to review our work. We greatly appreciate your insightful feedback and suggestions on how we can make our review better. We are particularly grateful for your advice regarding our tables and results. After carefully reviewing your feedback, we have implemented many of your suggestions. Please let us know if there is anything else we can do. 

Please find below our response:

Reviewer 3

Comment 1: Introduction. In the first sentence, the term "schizophrenia" is repeated twice. Please, review.

Thank you for your comment, we have adjusted the wording of the sentence so that our explanation of treatment-refractory schizophrenia is clearer. Page 1, Line 30-31

Comment 2:  Tables. Table 5. Characteristics of Included Studies. Perhaps the studies could be ordered by country and by year of publication.

Thank you for this feedback. We agree that by grouping the studies by country and ordering by the year of publication, Table 5 will appear more organised. We have made the necessary adjustments. Page 10-14, Table 5

Comment 3: Perhaps the number of studies and patients could be included in the title.

Thank you for your suggestion, we have added the number of studies and total participant number into our Table 5 title and main text. Page 10, Line 201

Comment 4: Also consider adding a column that includes the number of patients from each of the 23 studies.

Thank you for your comment. We have added a column specifying the number of clozapine users from each included study into the “Characteristics of Included Studies” table as per your recommendation. Please let us know if there are any further changes needed to our tables or if our interpretation of your advice is incorrect. Page 10-14, Table 5.

Comment 5: Maybe tables 5-7 could be merged.

Thank you for your feedback. We have merged Tables 6 and 7. Given the number of columns of information we would like to present; we decided to keep Table 5 separate. Page 16 Table 6.

Comment 6: Discussion. The strengths section is missing.

Thank you for your feedback. We agree that this is something that we have missed and should be explicitly mentioned. Please find our strength on Page 41, Lines 99-109.

Comment 7: References 100 and 111. Are they completed?

Thank you for your comment. We have updated our reference 100 (now 104) to include the journal page number. Page 49, Line 454. Reference 111 (Now reference 116) refers to the Australian Immunisation Handbook and has also been edited to conform with proper referencing formatting. Page 49, Line 484.
